# Anti-Tumor Potential of Post-Translational Modifications of PD-1

Xiaoming Xi  and Wuli Zhao *

State Key Laboratory of Respiratory Health and Multimorbidity, Institute of Medical Biotechnology, Chinese Academy of Medical Sciences, School of Basic Medicine Peking Union Medical College, Beijing 100005, China mingxiaoxi2017@163.com
* Correspondence: wenlyzh@imb.pumc.edu.cn

**Abstract:** Programmed cell death protein-1 (PD-1) is a vital immune checkpoint molecule. The location, stability, and protein–protein interaction of PD-1 are significantly influenced by post-translational modification (PTM) of proteins. The biological information of PD-1, including its gene and protein structures and the PD-1/PD-L1 signaling pathway, was briefly reviewed in this review. Additionally, recent research on PD-1 post-translational modification, including the study of ubiquitination, glycosylation, phosphorylation, and palmitoylation, was summarized, and research strategies for PD-1 PTM drugs were concluded. At present, only a part of PD-1/PD-L1 treated patients (35–45%) are benefited from immunotherapies, and novel strategies targeting PTM of PD-1/PD-L1 may be important for anti-PD-1/PD-L1 non-responders (poor responders).

**Keywords:** PD-1; immunotherapy; post-translational modification; ubiquitin–proteasome system; glycosylation; phosphorylation; palmitoylation

## 1. Introduction

The goal of cancer immunotherapy is to use the patient's immune system to thwart the tumor's progression [1]. In the tumor microenvironment (TME), which is intimately linked to the immunological escape of tumor cells, tumor-infiltrating lymphocytes have a regulatory function on the immune system [2]. One of the most popular immunotherapies is immune checkpoint blocking, particularly with the use of anti-PD-1 monoclonal antibodies. By obstructing immune checkpoint pathways, cancer cells can evade the immune system. Compared with chemotherapy drugs, immunotherapy can provide advantages such as personalization and long-lasting benefits [3]. At present, discovered immune checkpoint signaling molecules include programmed cell death protein-1 (PD-1), programmed death ligand-1 (PD-L1), cytotoxic T lymphocyte-associated antigen-4 (CTLA-4), etc. [4]. The PD-1/PD-L1 pathway is one of the immune checkpoints that is most crucial to the management of malignant tumors [5]. More than 1000 antibodies targeting the PD-1/PD-L1 pathway have been evaluated in clinical trials and are approved for use in a variety of malignancies, such as melanoma [6], non-small cell lung cancer [7], renal cell carcinoma, etc. [8]. Finding therapeutic approaches to increase the effectiveness of blocking immune checkpoints is critically needed. Even while the majority of patients who use PD-1/PD-L1 blockers see a significant improvement in their prognosis, long-lasting therapy benefits are also hard to obtain [9,10].

PD-1, also known as CD279, is a protein encoded by the 5-exon PDCD1 gene located on chromosome 2q37.3. It was first isolated by Ishida in 1992 using subtractive hybridization [11]. It consists of 288 amino acids and has a molecular weight of 31.6 kDa. PD-1 contains a single immunoglobulin V-like domain, a transmembrane domain, an immunoreceptor tyrosine-based inhibitory motif (ITIM) (V/L/I/XPYX/L/V), and an intracellular domain with the immunoreceptor tyrosine-based switch motif (ITSM) (TXpYXXV/I) [12,13]. In addition, studies have shown that PD-1 exists in two forms: membrane-bound and free. The plasma level of soluble PD-1 can predict the occurrence, prognosis, and treatment

effectiveness of cancer. PD-1 is primarily expressed as a transmembrane receptor in T-cells, B-cells, monocytes, and macrophages [14]. In vivo, PD-1 needs to be maintained at an appropriate level because its overexpression inhibits the activity of T-cell effectors and mediates tumor immune escape [15].

This review centers on the post-translational regulation of PD-1, encompassing ubiquitination, glycosylation, phosphorylation, palmitoylation, and palmitoylation, as well as drug development and existing challenges for PD-1 post-translational modification (PTM). It also emphasizes the utilization of these regulatory mechanisms in tumor immunotherapy (Table 1).

**Table 1.** Biological effects of post-translational modification of PD-1.

| PD-1 PTMs | Modification Sites | Related Enzymes | Biological Effects | References |
|---|---|---|---|---|
| Ubiquitination | K210, K233 | FBXO38, KLH22, c-Cbl, FBW7 | Reduce the expression of PD-1 | Refs. [15–17] |
| Glycosylation | N49, N58, N74, N116 | B3GNT2, FUT8 | Increase the affinity with PD-L1 and increase the stability of PD-1 | Refs. [18,19] |
| Phosphorylation | Y223, Y248 | SHP1, SHP2 | Inhibition of TCR signaling pathway, thereby inhibiting T-cell proliferation and activation | Ref. [20] |
| Palmitoylation | Cys192 | DHHC9 | Recycling endosome, thus preventing its lysosome-dependent degradation | Ref. [21] |

## 2. Overview of the PD-1/PD-L1 Checkpoint Pathway

Under typical physiological circumstances, the immune system can effectively surveil, accurately identify, and eliminate aberrant cells in a timely fashion to thwart the development of cancer. Nevertheless, cancer cells have the ability to evade the body's immune system by up-regulating or down-regulating specific molecules in order to attain immortality. Immune checkpoint therapy, also referred to as immune checkpoint blockade, has emerged as a compelling and promising therapeutic approach for numerous cancer patients. The first immune checkpoint inhibitor medication authorized by the FDA is called imilimumab. In recent years, the targeting of PD-1/PD-L1 has evolved as a novel strategy for cancer treatment. According to research statistics, as of 2020, there have been close to 5000 patents, with over 2000 focusing on the application of PD-1 in clinical trials and more than 1000 related to the application of PD-L1 antibodies in clinical trials. A variety of monoclonal antibodies have received approval from the FDA for the treatment of different types of cancer. These drugs encompass PD-1 targeted medications, including nivolumab, pembrolizumab, cemiplimab, toripalimab, and camrelizumab, as well as PD-L1 targeted medications, such as durvalumab and atezolizumab [22].

In the context of adaptive immunity, the activation of T-cells necessitates the coordinated action of two signals [23–25]. The initial signal occurs when the major histocompatibility complex (MHC) molecule of the antigen-presenting cells (APCs) presents the processed antigen to the T-cell receptor (TCR) of the T-cell, leading to the recognition of the antigen by the T-cell. The second signal is a non-antigen-dependent, costimulatory, or coinhibitory signal delivered by APCs [26,27]. This signal plays a crucial role in determining the fate of T-cells, influencing whether they differentiate into lethal effector cells or non-reactive cells. The molecules associated with the secondary signaling of T-cells, namely CD28, PD-1, and CTLA4, interact with B7-1/B7-2, PD-L1/PD-L2, and B7-1/B7-2, respectively [28]. Upon binding to its ligand, CD28 triggers cell cycle progression, interleukin-2 (IL-2) production, and clonal expansion [29]. In contrast, PD-1 and CTLA-4 elicit T-cell tolerance and inactivation upon binding to their specific ligands [12,30].

In order to maintain the proper immunological balance in the human body, the PD-1 receptor on cells normally interacts with its ligand, PD-L1, to produce negative regulatory effects that limit excessive proliferation and activation of T-cells, as shown in Figure 1. PD-1 has two binding ligands, PD-L1 and PD-L2 (with 34% homology). While PD-L2 exhibits a higher affinity for PD-1, the interaction between PD-1 and PD-L1 has a wider range of biological implications, with PD-L1 serving as the primary ligand for PD-1 [31]. Inhibiting the PD-1/PD-L1 pathway has the potential to reinstate T-cell functionality and effectively suppress Treg cells within the TME, thereby eliciting an anti-tumor immune response [32,33]. Co-stimulation or exogenous IL-2 has the potential to suppress the PD-1/PD-L1 pathway, with CD8+ T-cells exhibiting heightened sensitivity to this pathway as a result of their limited IL-2 production. Moreover, CD8+ T-cells are known to generate elevated levels of interferon-$\gamma$ (IFN-$\gamma$), which has the capacity to enhance the expression of PD-L1 in diverse cell types. The overexpression of PD-1 and its ligand PD-L1 can lead to the development of an immunosuppressive TME, enabling tumor cells to evade surveillance and destruction by the immune system. Inhibitors have the capacity to obstruct the PD-1/PD-L1 signaling pathway, thereby augmenting T-cell activation for the identification and elimination of tumor cells. This mechanism offers enduring survival advantages to individuals with cancer [34]. In 2002, Iwai et al. conducted animal experiments and discovered that the inhibition of the PD1/PD-L1 pathway resulted in a significant reduction in the growth of J558L myeloma cells in mice [35]. In 2005, it was confirmed that the deletion of PD-1 by PD-1 deletion transgenic mice resulted in enhanced effector T-cell function and promoted the accumulation of cytotoxic T-cells at the tumor site [36]. The antitumor effect exhibited prolonged duration.

The TCR initiates the adaptive immune response by identifying antigens displayed by tumor MHC class I. PD-L1 interacts with PD-1 and exerts an immunosuppressive function. The use of an anti-PD-1 antibody or anti-PD-L1 antibody can effectively inhibit immune dysfunction caused by the PD-1/PD-L1 axis, leading to the alleviation of T-cell suppression and the augmentation of the anti-tumor response.

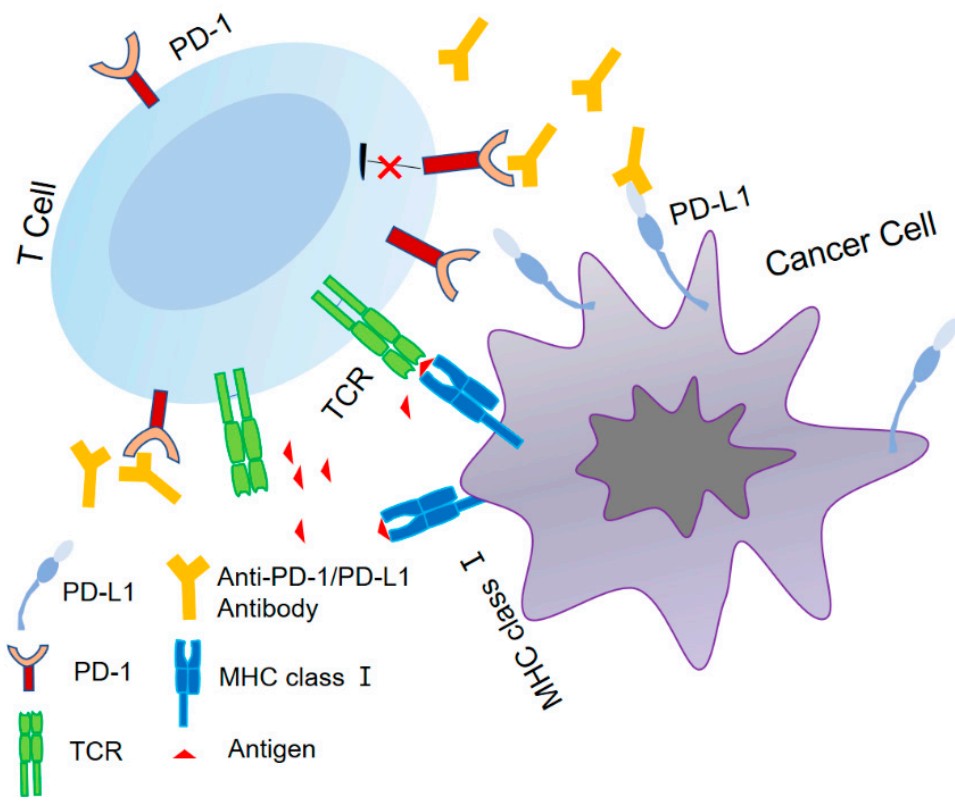

**Figure 1.** The identification of PD-1/PD-L1.

## 3. Ubiquitin–Proteasome System and Deubiquitination

Ubiquitin is a small protein characterized by a highly conserved sequence, consisting of a ubiquitin-activating enzyme (E1), a ubiquitin-conjugating enzyme (E2), and a ubiquitin-protein ligase (E3). A total of 1332 ubiquitin regulatory proteins have been identified in humans. Among these, E3 ligase and deubiquitin primarily participate in immune processes. The process of ubiquitin–protein ligation necessitates the sequential action of E1, E2, and E3 enzymes, which is crucial in facilitating protein degradation via endocytosis, signal transduction, receptor down-regulation, and other pathways [37,38]. The ubiquitin–proteasome pathway plays a role in various phases of PD-1 protein synthesis and transportation. Following the activation of T-cells, FBXO38, the E3 ligase of PD-1, facilitates lys48-linked polyubiquitination and subsequent proteasome degradation. The transcription levels of FBXO38 are decreased in tissues exhibiting rapid tumor progression, leading to elevated PD-1 expression on T-cells. Nevertheless, IL-2 has the capacity to significantly decrease the overexpression of PD-1, which is induced by the reduced transcription level of FBXO38 [16]. However, further investigation is needed to determine if other cytokines or signal transduction pathways play a role in the regulation of FBXO38 and PD-1. Another PD-1-related protein, KLHL22, serves as an adaptor for the Cul3-based E3 ligase, targeting incompletely glycosylated PD-1 for degradation, thereby enhancing tumor immunity. The content of the text is influenced by signals of T-cell activation [15]. C-cbl has been identified as a novel ubiquitin E3 ligase of PD-1. The C-terminal and cytoplasmic tail of PD-1 are involved in the down-regulation of PD-1 and the reduction of its surface expression through the ring finger function of C-cbl [17]. In the most recent study of PD-1 ubiquitin ligase, Liu et al. employed bioinformatics and biochemical research techniques to identify a new E3 ligase, FBW7, which targets PD-1 Lys233 residues at K48, facilitating the degradation of PD-1 and augmenting in vivo anti-tumor immunity [39].

Naruse et al. devised the small molecule-assisted shutoff degron system by leveraging the degradation properties of the ubiquitin–proteasome system, wherein E3 identifies the degron and ubiquitinates the target protein PD-1. PD-1 subsequently undergoes degradation, resulting in the formation of a small peptide consisting of approximately 10 amino acids. The rapid ubiquitination and subsequent degradation of PD-1 can be facilitated by the introduction of small molecular compounds. Following the removal of the compounds, PD-1 will undergo re-expression, thereby facilitating the regulation of drug dosage in the treatment of this disease and delineating the biological role of the target protein [40]. Bioinformatics analysis indicates that a higher degree of ubiquitination in vivo is associated with a more favorable response to anti-PD-1 treatment [41].

Nevertheless, inquiries persist regarding the influence of different ubiquitin ligases on the degradation of PD-1 and the correlation between signaling pathways and their respective effects, necessitating further investigation. The pathological role of USP12 in other forms of cancer, especially in the regulation of the TME, is not yet fully understood. Currently, research on deubiquitination primarily centers on PD-L1, with limited advancement in understanding the impact of deubiquitinase on the stability of PD-1 [42,43].

## 4. Glycosylation

One common PTM that is crucial to immune control, information transduction, and protein localization accuracy is glycosylation [44]. Widespread glycosylation in the extracellular domain of PD-1 is critical for giving targets for the creation of immunosuppressants, and it is necessary for preserving the stability and proper membrane expression of PD-1 [45]. One of the most prevalent types of protein glycosylation in eukaryotes is asparagine-linked glycosylation. Asparagine-linked residues are catalyzed to create N-glycosidic bonds with oligosaccharides in the presence of oligosaccharide transferase, followed by oligosaccharide translocation. This mechanism is frequently observed in the rough endoplasmic reticulum (ER). Proteins are degraded in the cytoplasm by the ER when there is an improper adjustment to glycosylation [22].

PD-1 is an inhibitory receptor that is heavily glycosylated [46]; the extracellular IgV domain contains four potential N-glycan sites, which are N49, N58, N74, and N116. Further, there are conserved domains that contain sites of N-linked glycosylation [47]. According to Sun et al., PD-1 was shown to be highly fucosylated and widely N-glycosylated in T-cells. The interaction between PD-1 and PD-L1 was revealed to be mediated by glycosylation at the N58 site. The structure of glycan would alter upon TCR activation [19]. In 2018, the FDA approved cemiplimab for the treatment of cutaneous squamous cell carcinoma based on the excellent results of clinical trials [48]. The antibody mostly binds to the BC and FG rings of PD-1, and cemiplimab's binding to PD-1 and its blocking impact on PD-L1 are greatly diminished in the absence of N-58 glycosylation [49]. Camrelizumab binds to the N58 polysaccharide of PD-1, and the loss of glycosylation causes a decrease in the affinity of PD-1 to PD-L1 [50]. Shi et al. reduced N74 glycosylation using an adenine base editor, which led to reduced PD-1 expression and attenuated the inhibition of CAR-T-cells [51]. The Asn58 residue of PD-1 is glycosylated with two N-acetyl glucosamines, one fucose, and two mannoses. Mw11-h317, a monoclonal antibody, stimulates T-cell immunity and suppresses tumor growth by targeting the glycosylation domain of PD-1ASN58. The interaction between PD-1 and MW11-H317 does not affect the conformation of PD-1. MW11-H317 demonstrated an overlapping binding region primarily in the FG loop of PD-1, in comparison to the target sites of nivolumab and pembrolizumab [52]. Previous studies have speculated that PD-1 glycosylation plays a role in the recognition of PD-1 in nivolumab. Tan et al. discovered, through surface plasmon resonance, that the PD-1 mutant and PD-1 wild type exhibited similar affinity. Moreover, the affinity of PD-1 expressed by Escherichia coli for nivolumab was similar to that of PD-1 expressed by mammalian cells (Kd = 1.45 nM). Therefore, the binding of nivolumab to PD-1 was not related to N-glycosylation, and N58 was the sole contact point at the interface of the PD-1–nivolumab–Fab complex. The PD-1 extracellular domain, specifically the N-loop,

plays a leading role [53]. Pembrolizumab interacts with the C'D ring of PD-1, although it has not yet been reported whether PD-1 is associated with PTM of pembrolizumab [54]. Similarly, the glycan modification of PD-1 had no significant effect on the binding of tori-palimab and tislelizumab [55,56]. Fucosylation is the process by which a specific fucosylase enzyme binds L-fucose to glycoproteins, glycolipids, or proteins, and it plays a crucial role in immune responses and the development of malignant tumors [57]. The loss of core fucosylation of PD-1 increases the ubiquitination of this protein, leading to its degradation in the proteasome and enhancing the activation of T-cells. This process effectively inhibits tumor growth. In TME, the dysregulation of PD-1 glycosylation (fucosylation, sialylation, N- and O-linked glycan branching, and O-glycan truncation) may result in the interaction of cemiplimab-rwlc, MW11-h317, and other antibodies with the N-58 site, thereby reducing their binding affinity with PD-1 [49,58].

## 5. Phosphorylation

Protein phosphorylation is essential for regulating cell proliferation, differentiation, protein conformational changes, and the transmission of signaling pathways. The process is tightly controlled by kinases and phosphatases, and it is reversible [59].

The Src homology-2 domain-containing protein tyrosine phosphatase (SHP) belongs to the subfamily of non-receptor tyrosine phosphatases, which is further categorized into SHP1 and SHP2. The phosphatase activity of SHP2 has a substantial effect on the phosphorylation of PD-1 and can dephosphorylate the tail of PD-1 as a component of a feedback mechanism [60,61]. In this process, PD-1 is activated by TCR, leading to the phosphorylation of tyrosine residues within ITIM and ITSM in its cytoplasmic tail. This activation leads to the recruitment of SHP-1, subsequently resulting in the dephosphorylation of CD28 and downstream signaling pathway molecules of TCR. The SHP2 protein exhibits widespread expression in T-cells and is positively associated with PD-1 expression in T-cells that have invaded tumors. Two tandem Src homology-2 (SH2) domains are present: the N-terminal (N-SH2) and C-terminal (C-SH2) [62]. Ultimately, they impede the activation, proliferation, and survival of T-cells [63]. Moreover, SHP2 has the capability to connect the phosphorylated PD-1 of two molecules via its N-terminal and C-terminus, thereby facilitating the formation of PD-1: PD-1 dimers within living cells [64]. Moreover, the interaction between SHP2 and PD-1 has an inhibitory impact on Th1 cells within the TME. PD-1 contains the ITIM domain at Y233 and the ITSM domain at Y248. The ITSM/ITIM plays a pivotal role in mediating the interaction between PD-1 and SHP2. The phosphorylation of PD-1 at tyrosine 248 serves as an indicator of the activation of the PD-1 pathway in T-cells. The phosphorylation of this site is crucial for PD-1 to exert its inhibitory function. The phosphorylated form of PD-1 can be employed in tumor biopsies to assess the activation status of this pathway. It can also function as an indicator of self-tolerance, whereas a decrease or loss of its expression may suggest autoimmunity [20]. It has been reported that concurrent phosphorylation of both the ITSM and ITIM of PD-1 results in the strong binding of the ITSM to C-SH2, leading to the recruitment of SHP2 to the proximity of PD-1. Concurrently, the ITIM binds to N-SH2, leading to its displacement from the catalytic pocket prior to the full activation of SHP2 [63,65,66]. The experimental findings by Patsoukis et al. demonstrated the association of SHP2 with ITSM-Y248 on two PD-1 molecules through C-SH2 and N-SH2 following PD-1 phosphorylation. Furthermore, it was observed that only ITSM-Y248 played a crucial role in the inhibitory function of PD-1 [67]. This phenomenon occurs when the concentration of PD-1 is sufficiently high, allowing ITSM to supplant ITIM in N-SH2 [68]. A variety of agents have been developed to treat tumors by targeting the pathway that inhibits the PD-1/SHP2 interaction [69–71]. In the context of Zymosan-induced inflammation, the PD-1 signaling pathway may lead to a decrease in the production of M1-type cytokines as a result of diminished phosphorylation of tyrosine residues in the PD-1 receptor/ligand and reduced recruitment of SHP2 by PD-1 [72].

## 6. Palmitoylation

Lipidation of proteins is considered one of the forms of PTM and is categorized as co-translational. Various types of fatty acyl groups attach to proteins, significantly contributing to protein localization, activation, structure, and stability [73]. Palmitate frequently accumulates in tumor cells, and an abnormal elevation in its concentration can lead to higher protein lipidation [74]. Palmitate is the most abundant fatty acid in cells (20%~30% of total fatty acids). Palmitoylation is a common form of fatty acylation. Forty years ago, Schmidt and Schlesinger first discovered a reversible PTM process called palmitoylation (S-acylation) [75]. This process involves attaching a 16-carbon palmitate to protein cysteine residues through thioester linkages (R-S-CO-R'). The process is regulated by palmitoyl acyltransferases containing zinc finger aspartate–histidine–histidine–cysteine motifs, and two groups of thioesterases are dynamically and reversibly regulated. The covalent binding of palmitate to membrane protein cysteine plays a significant role in regulating immune checkpoints [76,77].

The palmitoylation of PD-1 enhances its association with Rab11, a crucial signaling pathway for delivering PD-1 to recycling endosomes. This process also diminishes the degradation of the PD-1-dependent lysosomal pathway and aids in the relocation of PD-1 to the cell surface. The findings indicate that palmitoylation is a biochemical process catalyzed by the Asp–His–His–Cys (DHHC) acyltransferase family. An enzyme inhibitor has been developed to block the palmitoylation of PD-1, specifically targeting DHHC9. According to the results, the peptide demonstrated a dose-dependent inhibition of tumor growth in vitro. Research has shown that palmitoylation of PD-1 activates the mTOR signaling pathway, promoting tumor growth [21].

## 7. Research Strategies Targeting PD-1 PTM and Drug Development

As of 2022, ten monoclonal antibody medications have been authorized for clinical usage against PD-1; among these, nivolumab, pembrolizumab, toripalimab, and tislelizumab [55] mostly work by interacting with the structural elements of PD-1 to produce antibody-blocking effects. The glycosylation sites of PD-1 N-glycosylation are targeted by cemiplimab, penpulimab, and camrelizumab [78]. A domestic anti-PD-1 monoclonal antibody to IgG4 made using the yeast display approach is called sintilimab. The FG ring of PD-1 is home to the sintilimab/PD-1 complex epitopes, according to structural analysis [79]. A humanized IgG4 kappa monoclonal antibody called dostarlimab attaches itself to the PD-1 receptor and prevents it from interacting with PD-L1 and PD-L2 [80]. The effectiveness of dostarlimab, zimberelimab [81], and sintilimab in modulating the post-translational modification of PD-1 has not yet been described. Through hybridoma screening and antibody humanization technology, three compounds were found to disrupt the PD-1/PD-L1 pathway in vitro: MW11-h317, mAb059c, and STM418, which alter the N-58 glycosylation of PD-1. The safety and affinity of these tested antibodies are very good. A peptide inhibitor has been discovered to competitively block DHHC9, a significant PD-1 palmitoacylase interferer, with regard to palmitoacylation. Drug resistance brought on by SHP2 mutation or inactivation is the primary focus of the PTM of phosphorylation. The study of SHP2 allosteric inhibitors, such as the identification and refinement of SHP2 inhibitors PB17-026-01 and SHP099, is the primary focus in terms of therapeutic development direction [82,83]. The discovery and screening of E3 ligase, which has not been the subject of clinical studies, is the primary focus of research and development of PD-1 ubiquitination medications.

Bispecific antibodies are a class of drugs capable of simultaneously blocking two molecules, thereby enhancing the anti-tumor effect. Several bispecific antibodies currently under development for PD-1 include tebotelimab, which specifically targets PD-1 and LAG-3. This antibody shows potential for the treatment of advanced HER2+ gastric/gastroesophageal junction adenocarcinoma. Cadonilimab, a medication targeting PD-1 and CTLA-4, demonstrates efficacy in treating various types of cancer, including cervical cancer, lung cancer, gastric/gastroesophageal junction cancer, and nasopharyngeal can-

cer [84,85]. Several other bi-specific antibodies in clinical trials include MEDI5752 (PD-1 × CTLA-4), IBI319 (CD137 × PD-1), LY3434172 (PD-1 × PD-L1), MGD019 (PD-1 and CTLA-4), and CDX-527 (PD-1 × CD27) [86–91]. Xiong et al. have formulated a bi-specific antibody designed to target VEGF165 and PD-1, thereby creating a combined anti-angiogenic and immune checkpoint-blocking therapy [91]. The administration of anti-PD-1 has been shown to enhance the sensitivity of tumors to anti-angiogenic therapy, thereby extending its effectiveness in models of metastatic breast cancer, pancreatic neuroendocrine tumors, and melanoma. This strategy enhanced the proliferation of T-cells and preserved T-cell activation. Furthermore, bi-specific antibodies have been created to target PD-1 × GITR-L, TIM3 × anti-PD-1, and c-Met × PD-1, and their safety, efficacy, and other factors still require verification through clinical trials [92–94].

Furthermore, a new therapeutic strategy called precision medicine entails creating a personalized treatment plan that is specific to each patient's circumstances. Precision medicine can be advanced in the treatment of head and neck cancer by using genomic and multi-omics mapping [95]. Better treatment outcomes can be achieved by choosing the right immunotherapy, claims the precision medicine initiative [96,97].

## 8. Conclusions and Future Perspectives

Even though immune checkpoint-blocking medication has been quite effective in clinical settings, most patients find it difficult to see long-term therapeutic results. Although the influence of PD-1 PTM on tumor treatment resistance is significant, it is now primarily studied in relation to SHP2 inhibitors [42,98]. SHP2, an enzyme that is essential for phosphorylating PD-1, is a possible target for anticancer medications. Allosteric inhibitors of SHP2 are the subject of numerous clinical trials at the moment, and SHP2 mutations can cause drug resistance in cancer cells and reduce the effectiveness of targeted tumor therapy. The dysregulation of ubiquitination and deubiquitination processes is implicated in the pathogenesis of tumors. The decreased or inactivated expression of deubiquitination enzymes in cancer can result in the proliferation and drug resistance of tumor cells. The PTM of the deubiquitination enzyme in PD-1 has not been documented in the existing literature. Yang et al. discovered that the decreased expression of the deubiquitination enzyme USP12 resulted in the reduced sensitivity of murine lung tumor cells to PD-1 blockade therapy, leading to enhanced resistance to drugs [42]. The literature indicates that USP12 has been shown to diminish T-cell activation by facilitating the expression of specific chemokines. However, the impact of USP12 on PTM of PD-1 is not addressed in this paper, despite its potential significance for future investigation. Tumor resistance to PD-1 treatment can be categorized into internal and external factors. Based on the literature review, internal factors primarily stem from impediments in tumor immune recognition, epigenetic regulation, abnormal carcinogenic signaling, and the IFN-γ signaling pathway. However, PTM plays a crucial role in the structure of eukaryotic proteins. Given the insufficient attention to the regulation of tumor drug resistance through the PTM of PD-1, there is a need for fundamental and exploratory research on targeting the PTM of PD-1 to impede the progression of tumor drug resistance. This review also presents novel concepts regarding tumor resistance to immunotherapy.

It is important to highlight that a range of therapies have been devised for the PTM of PD-L1, including anti-gPD-L1-MMAE, CDK4/6 inhibitor+anti-PD-1, etc. [99,100]. Research on the PTM of PD-L1 is more extensive and comprehensive in comparison to PD-1. For instance, glycosylation at the glycosylation sites N192, N200, and N219 of PD-L1 prolongs its half-life by 4 h, whereas non-glycosylated PD-L1 is phosphorylated by glycogen synthase kinase 3β and subsequently degraded by ubiquitin/proteasome [101]. Furthermore, PD-L1 exhibits two mechanisms of mono-ubiquitination and multi-ubiquitination [102,103]. Nevertheless, recent research has initiated investigations into the PTM of PD-1. However, these studies have primarily concentrated on specific aspects, such as ubiquitination and glycosylation, leaving numerous unresolved issues that warrant further exploration. For instance, despite reports indicating that PD-1 can induce polyubiquitin via the E3 ligase,

it remains uncertain whether there exists a deubiquitination enzyme to counteract the ubiquitination process of PD-1. Acetylation of ubiquitin protein can lead to antagonism, thereby governing protein stability and subcellular localization. Research on acetylation has predominantly concentrated on histone acetylation at the transcriptional level, while the issue of PD-1 at the protein level also warrants exploration with respect to acetylation regulation.

In cancer, epigenetic modifications silence chemokines such as CXCL9 and CXCL10 in the TME, enabling tumor immune evasion [104]. Gene mutations, such as the KRAS-G12D mutation and isocitrate dehydrogenase-1, can result in resistance to PD-1/PD-L1 immunotherapy in patients [105,106]. Activation of the Wnt-β-catenin signaling pathway may lead to immune tolerance [107]. As a result, the majority of patients are unable to benefit from immune checkpoint blockade therapy. This article summarizes and discusses therapeutic strategies for PTM of PD-1. These findings may pave the way for new applications of immunotherapy and suggest new avenues for future research.

Further research is needed to focus on the post-translational modification of PD-1, which holds significant importance in understanding the role of PD-1 in T-cells and the interaction of PD-1/PD-L1. In the future, researchers should prioritize exploring drug combinations, such as bifunctional antibodies, and refining and regulating preclinical models to improve the effectiveness of precision medicine in response to the complex nature of the TME.

**Author Contributions:** Conceptualization, W.Z.; writing—original draft preparation, X.X.; writing—review and editing, W.Z. and X.X. All authors have read and agreed to the published version of the manuscript.

**Funding:** This work was supported by grants from the Chinese Academy of Medical Sciences Innovation Fund for Medical Sciences (2021-I2M-1-030, 2023-I2M-2-001 China).

**Institutional Review Board Statement:** Not applicable.

**Informed Consent Statement:** Not applicable.

**Data Availability Statement:** Not applicable.

**Conflicts of Interest:** The authors declare no conflicts of interest.

## Abbreviations

| | |
|---|---|
| PD-1 | Programmed cell death protein-1 |
| PD-L1 | Programmed death ligand-1 |
| CTLA-4 | Cytotoxic T lymphocyte-associated antigen-4 |
| TME | Tumor microenvironment |
| ITIM | Immunoreceptor tyrosine-based inhibitory motif |
| ITSM | Intracellular domain of immunoreceptor tyrosine-based switch motif |
| MHC | Major histocompatibility complex |
| APCs | Antigen-presenting cells |
| IL-2 | Interleukin-2 |
| IFN-γ | Interferon-γ |
| N-SH2 | N-terminal |
| C-SH2 | C-terminal |
| SH2 | Src homology-2 |
| DHHC | Asp-His-His-Cys |
| E1 | Ubiquitin-activating enzyme |
| E2 | Ubiquitin-conjugating enzyme |
| E3 | Ubiquitin-protein ligase |
| ER | Endoplasmic reticulum |
| SHP | Src homology-2 domain-containing protein tyrosine phosphatase |
| PTM | Post-translational modification |

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
