# Peer review of "Anti-Tumor Potential of Post-Translational Modifications of PD-1"

_cimb, doi:10.3390/cimb46030136_

Round 1

Reviewer 1 Report

Comments and Suggestions for Authors

The authors have prepared a review on a rather interesting topic on post-translational modifications of PD1.

However, there are a lot of similar papers, within the research topic:

https://www.frontiersin.org/articles/10.3389/fimmu.2023.1230135/full

https://www.ncbi.nlm.nih.gov/pmc/articles/PMC6242346/

https://www.sciencedirect.com/science/article/pii/S1044579X21001036

https://onlinelibrary.wiley.com/doi/full/10.1111/imm.13573

https://www.ncbi.nlm.nih.gov/pmc/articles/PMC8615371/

https://www.nature.com/articles/s41388-018-0303-3

1) What is the novelty of this research?

2) I think that the authors should reorganize this MS and add a Section: "Research strategies targeting PD-1 post-translational modification and drug development." (to extract this information from conclusions section)

3) It is important to clearly discuss in the main text and in the abstract that only a part of PD-1/PDL1 treated patients (35-45%) are benefited from immunotherapies and novel strategies targeting PTM of PD-1/PDL1 may be important for anti-PD-1/PDL1 non-responders (poor responders).

Author Response

1) What is the novelty of this research?

Answer:1.As for the post-translational modification of PD-1/PD-L1, most of them are for the post-translational modification of PD-L1, just like the six links you sent, four of them only talk about the post-translational modification of PD-L1(https://www.frontiersin.org/articles/10.3389/fimmu.2023.1230135/full;https://www.ncbi.nlm.nih.gov/pmc/articles/PMC6242346/;https://www.ncbi.nlm.nih.gov/pmc/articles/PMC8615371/;https://www.nature.com/articles/s41388-018-0303-3

),While this link (https://www.sciencedirect.com/science/article/pii/S1044579X21001036

) tells the story of PD-1 only of ubiquitin and glycosylation, finally a link (https://onlinelibrary.wiley.com/doi/full/10.1111/imm.13573) tells the story of ubiquitin PD-1, glycosylation and phosphorylation, It can be seen that the current understanding of PD-1 post-translational modification is not as comprehensive as that of PD-L1. However, our review describes the ubiquitination and deubiquitination, glycosylation, phosphorylation and palmitoylation of PD-1.

  1. This review introduces drugs targeting PD-1 post-translational modification and current issues related to drug resistance at PD-1 post-translational modification sites, which can provide a reference for emerging precision medicine.
  2. the early work mainly focused on epigenetic, transcriptional and post-transcriptional regulation of PD-1/PD-L1. The current post-translational regulation of PD-L1 is more, therefore, this article provides new ideas for tumor immunotherapy.

  • I think that the authors should reorganize this MS and add a Section: "Research strategies targeting PD-1 post-translational modification and drug development." (to extract this information from conclusions section)

Answer:Thanks to your suggestion, we have added a seventh section to the article: "Research strategies targeting PD-1 post-translational modification and drug development"

7.Research strategies targeting PD-1 post-translational modification and drug development

As of 2022, ten monoclonal antibody medications have been authorized for clinical usage against PD-1; among these, nivolumab, pembrolizumab, toripalimab, and tislelizumab[58] mostly work by interacting with the structural elements of PD-1 to produce antibody-blocking effects. The glycosylation sites of PD-1 N-glycosylation are targeted by cemiplimab, penpulimab, and camrelizumab[81]. A domestic anti-PD-1 monoclonal antibody to IgG4 made using the yeast display approach is called sintilimab. The FG ring of PD-1 is home to the Sintilimab/PD-1 complex epitopes, according to structural analysis[82]. A humanized IgG4 kappa monoclonal antibody called dostarlimab attaches itself to the PD-1 receptor and prevents it from interacting with PD-L1 and PD-L2[83]. The effectiveness of dostarlimab, zimberelimab[84] , and sintilimab in modulating the post-translational modification of PD-1 has not yet been described. Through hybridoma screening and antibody humanization technology, three compounds were found to disrupt the PD-1/PD-L1 pathway in vitro: MW11-h317, mAb059c, and STM418, which alter the N-58 glycosylation of PD-1. The safety and affinity of these tested antibodies are very good. A peptide inhibitor has been discovered to competitively block DHHC9, a significant PD-1 palmitoacylase interferer, with regard to palmitoacylation. Drug resistance brought on by SHP2 mutation or inactivation is the primary focus of the post-translational modification of phosphorylation. The study of SHP2 allosteric inhibitors, such as the identification and refinement of SHP2 inhibitors PB17-026-01 and SHP099, is the primary focus in terms of therapeutic development direction[85, 86]. The discovery and screening of E3 ligase, which has not been the subject of clinical studies, is the primary focus of research and development of PD-1 ubiquitination medications.

Bispecific antibodies are a class of drugs capable of simultaneously blocking two molecules, thereby enhancing the anti-tumor effect. Several bispecific antibodies currently under development for PD-1 include Tebotelimab, which specifically targets PD-1 and LAG-3. This antibody shows potential for the treatment of advanced HER2+ gastric/gastroesophageal junction adenocarcinoma. Cadonilimab, a medication targeting PD-1 and CTLA-4, demonstrates efficacy in treating various types of cancer including cervical cancer, lung cancer, gastric/gastroesophageal junction cancer, and nasopharyngeal cancer[87, 88]. Several other bispecific antibodies in clinical trials, including MEDI5752 (PD-1×CTLA-4), IBI319 (CD137×PD-1), LY3434172 (PD-1×PD-L1), MGD019 (PD-1 and CTLA-4), and CDX-527 (PD-1×CD27)[89-93]. Xiong et al. have formulated a bispecific antibody designed to target VEGF165 and PD-1, thereby creating a combined anti-angiogenic and immune checkpoint blocking therapy[94]. The administration of Anti-PD-1 has been shown to enhance the sensitivity of tumors to anti-angiogenic therapy, thereby extending its effectiveness in models of metastatic breast cancer, pancreatic neuroendocrine tumors, and melanoma. This strategy enhanced the proliferation of T cells and preserved T cell activation. Furthermore, bi-specific antibodies have been created to target PD-1×GITR-L, TIM3×anti-PD-1, and c-Met×PD-1, and their safety, efficacy, and other factors still require verification through clinical trials[95-97].

  • It is important to clearly discuss in the main text and in the abstract that only a part of PD-1/PDL1 treated patients (35-45%) are benefited from immunotherapies and novel strategies targeting PTM of PD-1/PDL1 may be important for anti-PD-1/PDL1 non-responders (poor responders)

Answer:Thanks to your suggestion, we have highlighted the importance of PD-1 post-translational modifications in patients who do not respond to PD-1/PD-L1 immunotherapy in the summary and discussion sections.   

In cancer, epigenetic modifications silence chemokines such as CXCL9 and CXCL10 in the tumor microenvironment, enabling tumor immune evasion. Gene mutations, such as the KRAS-G12D mutation and isocitrate dehydrogenase-1, can result in resistance to PD-1/PD-L1 immunotherapy in patients. Activation of the Wnt-β-catenin signaling pathway may lead to immune tolerance. As a result, the majority of patients are unable to benefit from immune checkpoint blockade therapy. This article summarizes and discusses therapeutic strategies for post-translational modification of PD-1. These findings may pave the way for new applications of immunotherapy and suggest new avenues for future research.

Reviewer 2 Report

Comments and Suggestions for Authors

Very minor English corrections. I would just read over the paper and smooth some of the sections that had run-on/long lists so that its easier to read.

Additionally, there were some technical issues, I don't know if it was just my reviewer copy but line 70 "the FDA for cancer therapy" I have a teal painted object right after that sentence that should be removed if it's real.

Line 88: "In contrast, PD-1 and CTLA-88 4 elicit T-cell tolerance upon binding to their specific ligands"; I would insert anergy or inactivation in addition to tolerance with an appropriate reference.

Line 90: PD-1 is a cell surface receptor, here you say receptor on tissue which is confusing.

Comments on the Quality of English Language

Just very minor word choice appearances here and there.

Author Response

Very minor English corrections. I would just read over the paper and smooth some of the sections that had run-on/long lists so that its easier to read.

Answer:Thank you very much for your comments, we have carefully checked the text and some paragraphs have been streamlined.

Additionally, there were some technical issues, I don't know if it was just my reviewer copy but line 70 "the FDA for cancer therapy" I have a teal painted object right after that sentence that should be removed if it's real.

 Answer:Thanks for your comments and suggestions, we have revised the above sentence:

The first immune checkpoint inhibitor medication authorized by the FDA is called imilimumab.

Line 88: "In contrast, PD-1 and CTLA-88 4 elicit T-cell tolerance upon binding to their specific ligands"; I would insert anergy or inactivation in addition to tolerance with an appropriate reference.

Answer: We have added sentences and looked for references as a result of your recommendation: In contrast, PD-1 and CTLA-4 elicit T-cell tolerance and inactivation upon binding to their specific ligands[12,30].

  1. Ok CY, Young KH. Targeting the programmed death-1 pathway in lymphoid neoplasms [J]. Cancer Treat Rev, 2017, 54: 99-109.
  2. Probst HC, Mccoy K, Okazaki T, et al. Resting dendritic cells induce peripheral CD8+ T cell tolerance through PD-1 and CTLA-4 [J]. Nat Immunol, 2005, 6(3): 280-286.

Line 90: PD-1 is a cell surface receptor, here you say receptor on tissue which is confusing.

Answer: Thank you for your comments and suggestions. We have revised the original sentence: In order to maintain the proper immunological balance in the human body, the PD-1 receptor on cells normally interacts with its ligand, PD-L1, to produce negative regulatory effects that limit excessive proliferation and activation of T cells, as shown in Figure 1.

Comments on the Quality of English Language

Just very minor word choice appearances here and there.

Answer:Thank you for your suggestions, we have carefully checked and corrected the grammar errors in the review.

Reviewer 3 Report

Comments and Suggestions for Authors

The authors of the present work described the post-translational modifications of PD-1 as target for cancer therapies. 

The manuscript looks like well written and organized. The authors have presented an interesting topic in the field of cancer treatments. The paper should be considered after major revisions.

1.       In the first part of the manuscript the authors should add a short overview of the importance of immune infiltrate and TME in immunotherapies response;

2.       The topic of personalized medicine is not mentioned in the manuscript. This therapeutic approach represents an important challenge of modern medicine and the authors should better described the last innovation in this field. The following references should be included in the manuscript: “Precision medicine in Head and Neck Cancers: Genomic and Preclinical approaches. Doi: 10.3390/jpm12060854” and “Delivering precision oncology to patients with cancer. doi: 10.1038/s41591-022-01717-2”.

Author Response

  1. In the first part of the manuscript the authors should add a short overview of the importance of immune infiltrate and TME in immunotherapies response;

Answer: Thank you for your comments and suggestions. We have added to the first paragraph the importance of immune invasion and the tumor microenvironment:

The goal of cancer immunotherapy is to use the patient's immune system to thwart the tumor's progression[1].In the tumor microenvironment (TME), which is intimately linked to the immunological escape of tumor cells, tumor infiltrating lymphocytes have a regulatory function on the immune system. [2]

[1]A guide to cancer immunotherapy: from T cell basic science to clinical practice

[2]Cancer immunoediting: from immunosurveillance to tumor escape

The goal of cancer immunotherapy is to use the patient's immune system to thwart the tumor's progression. T cells are the foundation of the anti-tumor immune response and have the capacity to precisely kill tumor cells, whereas tumor infiltrating lymphocytes play an immunological regulatory role in the tumor microenvironment (TME) and are strongly associated to the immune escape of tumor cells.

  1. The topic of personalized medicine is not mentioned in the manuscript. This therapeutic approach represents an important challenge of modern medicine and the authors should better described the last innovation in this field. The following references should be included in the manuscript: “Precision medicine in Head and Neck Cancers: Genomic and Preclinical approaches. Doi: 10.3390/jpm12060854” and “Delivering precision oncology to patients with cancer. doi: 10.1038/s41591-022-01717-2”.

Answer:Thanks to your suggestions, we have added a description of precision medicine in the seventh part of the article.

Furthermore, a new therapeutic strategy called precision medicine entails creating a personalized treatment plan that is specific to each patient's circumstances. Precision medicine can be advanced in the treatment of head and neck cancer by using genomic and multi-omics mapping[98]. Better treatment outcomes can be achieved by choosing the right immunotherapy, claims the precision medicine initiative[99, 100].

Round 2

Reviewer 1 Report

Comments and Suggestions for Authors Dear Authors and Editor, I`m generally satisfied with the authors'
responses to my comments.

Line 65 - Please check the name of drug "The first immune checkpoint inhibitor medication authorized by the FDA is 65 called imilimumab". I think it is iPilimumab

Reviewer 3 Report

Comments and Suggestions for Authors

Now the manuscript is acceptable in the present form for the pubblication on CIMB journal